## Perspective

environmental science/fluid mechanics/ health and disease and epidemiology

COVID-19, makeshift hospitals, ventilation, natural ventilation, mechanical ventilation, airborne disease control

**Author for correspondence:**
P. F. Linden
e-mail: pfl4@cam.ac.uk

# Displacement ventilation: a viable ventilation strategy for makeshift hospitals and public buildings to contain COVID-19 and other airborne diseases

## Rajesh K. Bhagat and P. F. Linden

Department of Applied Mathematics and Theoretical Physics, University of Cambridge, Wilberforce Road, Cambridge CB3 0WA, UK

RKB, 0000-0002-8928-4534; PFL, 0000-0002-8511-2241

The SARS-CoV-2 virus has so far infected more than 31 million people around the world, and its impact is being felt by all. Patients with diseases such as COVID-19 should ideally be treated in negative pressure isolation rooms. However, due to the overwhelming demand for hospital beds, patients have been treated in general wards, hospital corridors and makeshift hospitals. Adequate building ventilation in hospitals and public spaces is a crucial factor to contain the disease (Escombe *et al.* 2007 *PLoS Med.* **4**; Escombe *et al.* 2019 *BMC Infect. Dis.* **19**, 88 (doi:10.1186/s12879-019-3717-9); Morawska & Milton 2020 *Clin. Infect. Dis.* ciaa939. (doi:10.1093/cid/ciaa939)), to exit lockdown safely, and reduce the chance of subsequent waves of outbreaks. A recently reported air-conditioner-induced COVID-19 outbreak caused by an asymptomatic patient, in a restaurant in Guangzhou, China (Lu *et al.* 2020 *Emerg. Infect. Dis.* **26**) exposes our vulnerability to future outbreaks linked to ventilation in public spaces. We argue that *displacement ventilation* (either mechanical or natural ventilation), where air intakes are at low level and extracts are at high level, is a viable alternative to negative pressure isolation rooms, which are often not available on site in hospital wards and makeshift hospitals. Displacement ventilation produces negative pressure at the occupant level, which draws fresh air from outdoors, and positive pressure near the ceiling, which expels the hot and contaminated air out. We acknowledge that, in both developed and developing countries, many modern large structures lack the openings required for natural ventilation. This lack of openings can be supplemented by installing extract fans. We

have also discussed and addressed the issue of the 'lock-up effect'. We provide guidelines for such mechanically assisted, naturally ventilated makeshift hospitals.

# 1. Introduction

COVID-19, the disease caused by the virus, SARS-CoV-2 is a pandemic and a global emergency which has stressed and in some cases overwhelmed healthcare systems across the world [1]. As of 23 September 2020, more than 31 million confirmed cases and over 977 000 fatalities have been reported, and these numbers are expected to increase significantly. Originating in Wuhan in China, it has rapidly spread to most of the world, and the current epicentres of the disease are in Asia, the USA and South America. Significant outbreaks of the disease have occurred in high population density, low income/developing countries such as India, Pakistan and Nigeria, and the disease is expected to cause significant numbers of fatalities in these countries as well.

The proliferation of the disease has overwhelmed many existing healthcare systems. In response to the overwhelming demand for hospital beds for critically ill patients, a new healthcare facility/hospital has been built in record time in Wuhan. Similar healthcare preparedness is imperative in both the developed and the resource-scarce developing world. In London, an exhibition and convention centre was converted into a 4000-bed Nightingale-style emergency hospital in 9 days. Other such new emergency hospitals have been created in the UK in Manchester, Birmingham, Bristol and Glasgow. In New York, makeshift tent-hospitals have been created in Central Park, and the Manhattan convention centre has been converted into a makeshift hospital. Suspected COVID-19 patients are being quarantined in hotels, and in India railway coaches are being converted to accommodate the patients and suspected cases.

COVID-19 is an infectious disease caused when the SARS-CoV-2 virus impacts on receptors in the body, usually in the respiratory tract. While the possibility of airborne infection transmission of the virus remains controversial [2,3], the virus can remain stable in aerosol form for hours [4], and can still potentially infect people [1,5,6]. In 2003, a possible SARS-CoV infection of healthcare workers during an aerosol-generating medical procedure had been reported [7]. Furthermore, Yu *et al.* [8] presented the evidence of long-distance airborne SARS contagion transport between the apartment blocks in Amoy Gardens housing complex in Hong Kong. More recently, an outbreak of cases in a restaurant in Guanghzou, China, where CCTV ruled out the possibility of fomite transmission provides further evidence of airborne transmission [9].

Modern medical as well as residential, commercial and industrial buildings are tightly constructed. To maintain an acceptable indoor environment (air quality and thermal comfort), pollutants and heat produced by human activities, appliances, machines, as well as by indoor commercial and industrial activities, need to be removed. However, partly in response to the energy crisis in 1973 and increasing concerns since then about climate change, until now the emphasis on design criteria for building ventilation has primarily been based on energy efficiency, and sustainability and thermal comfort. Only recently has the removal of indoor pollutants started to be a major consideration.

However, with the current pandemic, the focus has shifted and the primary objective is to minimize exposure of indoor contagion/bio-aerosol produced by an individual to other occupants present in the room, while also maintaining the thermal comfort.

Conventionally, in hospitals, patients with infectious respiratory diseases are kept in negative pressure isolation rooms, which provide ventilation to control the spread of the disease. However, in the current pandemic patients have been kept in the hospital wards and beds which were not intended to handle such cases. In this paper, we show that *displacement ventilation*, either natural or mechanical, where air is drawn in through lower-level inlets and extracted at high level, can significantly reduce the spread of airborne contagion, and save the lives of healthcare workers, carers and patients [2].

# 2. Building ventilation in healthcare settings

In a healthcare setting, the purpose of ventilation has been to remove harmful and unwanted agents—gases, pollutants, heat and contagion from the building envelope by indoor–outdoor exchange of air. There are two basic ventilation types (figure 1); *mixing ventilation* where the air is circulated throughout the whole space, for example, by ceiling fans or cool air introduced from high-level air

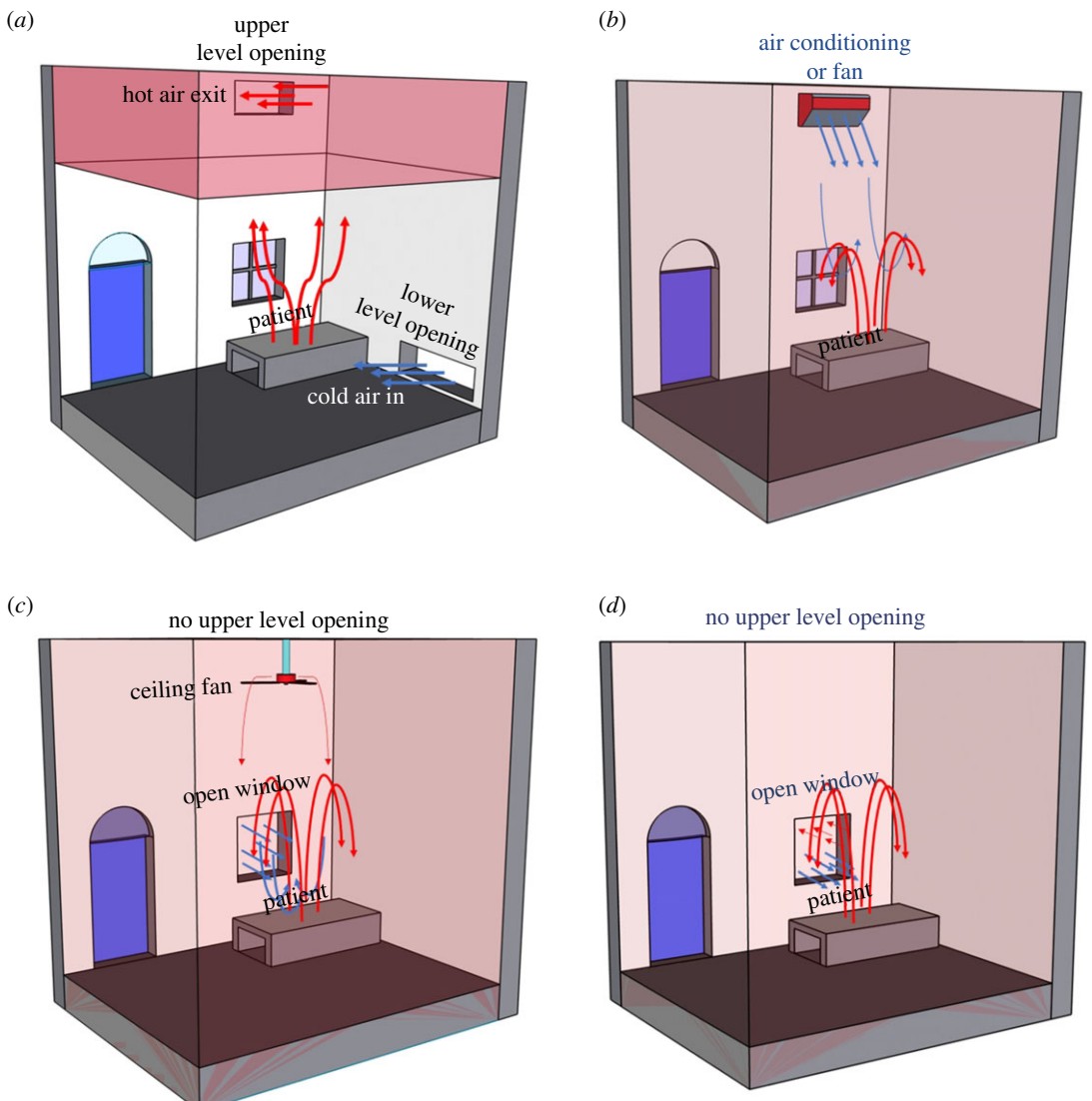

**Figure 1.** Schematic of different ventilation types, (*a*) displacement ventilation, (*b*) positive pressure mixing ventilation by air-conditioning, (*c*) positive pressure mixing ventilation by fan and open window, and (*d*) wind-driven mixing ventilation. In displacement ventilation, fresh air enters from the bottom of the room and contaminated air exits from the top. In mixing ventilation with a ceiling fan or high-level air conditioning unit, the contaminant/contagion is distributed uniformly throughout the whole room. These ventilation modes can either be produced mechanically by fans or air conditioning, or by using natural ventilation.

conditioning ducts with a view to maintaining uniform conditions such as temperature throughout the space, and *displacement ventilation*, where fresh air is introduced through lower-level inlets and extracted at high level in the space. Of these two modes mixing ventilation is the most commonly used, both in mechanical and natural ventilation.

Displacement ventilation relies on the fact that inevitably heat is generated within the space, either from the occupants and/or equipment, which produces hot air, which rises and accumulates near the ceiling. Consequently, the ambient conditions experienced by occupants in these two different ventilation modes are fundamentally different. In mixing ventilation, the occupant is surrounded by mixed air of uniform properties, such as the concentration of a contaminant—a bit like lying in dirty bath water. By contrast, in displacement ventilation the occupant is surrounded by newly arrived air, and pollutants generated near or by the person are lifted towards the ceiling by the rising warm air—rather like being in an inverted shower and rinsed with clean water. Of course, for the same amount of contaminant generation and ventilation flow, the total amount of contaminant in the space is the same: in mixing ventilation, it is everywhere at a relatively low concentration, while in displacement ventilation the contaminant is concentrated in a warm layer near the ceiling. When the system is

designed correctly this contaminated layer is located above the occupants and so they are only exposed to clean uncontaminated air.

Consequently, displacement ventilation has the potential to provide much better local conditions for patients within hospitals. How is this achieved? Essentially fresh air needs to be introduced at lower levels in the ward so that it flows over the patient collecting air expelled from their lungs and heat released from their body and any surrounding equipment. This air is then warmer and rises above the patient and accumulates near the ceiling where it needs to be extracted from the building.

As a consequence of this efficient flow pattern, the ventilation rate, i.e. the rate at which air is exchanged between inside and outside a space, required for a healthy environment in displacement ventilation is much less than that required for mixing ventilation. For contagious disease hospital wards, the World Health Organization (WHO) recommends—apparently based on mixing ventilation—a ventilation rate of 160 LSP (litres per second per person). This is much greater than the 10 LSP of fresh air recommended for breathing [10], and would require 16 air changes per hour (ACH) in a typical cubicle measuring 4 m wide, 3 m long and 3 m high.

For patients with an airborne contagious disease, a negative pressure ventilated isolation room is ideal, as it does not allow contagion to escape from the isolation room. However, building and running new isolation rooms requires time, resources and maintenance staff, which in the current pandemic, and in many societies, are scarce. To contain the disease, displacement ventilation in makeshift hospitals and other buildings can be a viable and safe alternative.

Mixing ventilation, as the term implies, mixes the air within a space. This implies that aerosols and other contaminants are mixed throughout the space so that all occupants are exposed to them. In the case of COVID-19, the virus is transported on droplets expelled from an infected person. The exact nature of this transport depends on the amount and size distribution of droplets, and whether the individual is breathing, talking, coughing or sneezing. We assume that in hospitals all persons are wearing personal protective equipment (PPE) so that the impact of coughing and sneezing are greatly reduced. In any event, a person coughing every minute for 2 h produces about 150 l of exhaled air, while simple breathing or talking produce 10 times that amount over the same period [11]. So even in the absence of PPE, while low-frequency intermittent events such as coughing and sneezing produce large quantities of droplet per event, under normal circumstances breathing and speaking account for most of the bio-aerosol produced during the day [12–16]. Indeed, in enclosed spaces such as hospital wards mixing caused by air conditioning or ceiling fans spreads the contamination throughout the space, and so makes the '1–2 m' social distancing rule problematic [3]. Instead we recommend that displacement ventilation be used to provide a safe environment for occupants in a building. We first discuss displacement ventilation in the context of natural ventilation, and subsequently consider the effects of mechanical extraction by ceiling fans.

# 3. Natural ventilation

Both modes of ventilation shown in figure 1 can be produced either mechanically or by natural ventilation. Natural ventilation relies on free natural resources, temperature differences between inside and outside the building—known as 'stack-driven' ventilation and the wind, to drive flow through the building. As was recognized by Florence Nightingale [17], there is evidence that natural ventilation can be beneficial for patient health even when mechanical ventilation is available [18,19], and in many cases it will be the only option.

## 3.1. Wind-driven natural ventilation

Wind-driven ventilation depends upon wind speed and direction and turbulence intensity, all of which fluctuate in time. Urban layout, building type, building location, the size and density distribution of the surrounding buildings, and the urban heat island, make it extremely challenging to predict the ventilation rate. Furthermore, the fluctuating indoor flows caused by wind-driven ventilation may produce random distribution of the unwanted agents, which in the context of removal of the airborne contagion from patients is undesirable. However, when openings are appropriately arranged, for example by ensuring that roof extracts are at negative wind pressure, wind-driven ventilation can assist displacement ventilation and make it more effective [20]. Consequently, in the following we ignore the effects of wind and concentrate on stack-driven ventilation, noting that care must be taken to ensure that any wind-driven flow assists, and does not disrupt, the flow patterns established by temperature differences.

## 3.2. Displacement ventilation or stack-driven natural ventilation

In stack-driven natural ventilation warm, buoyant air (due to body heat or the heat generated by solar gains, equipment and appliances) rises towards the ceiling and exits through an upper-level opening. This, in turn, draws in cooler (higher density) outdoor air that flows across the floor of the room. The stratification produced by the indoor temperature gradient drives the flow inside the building (figure 1a). The flow is predominantly unidirectional, upwards from the floor, removing airborne contagion away from patients and other occupants towards the ceiling where it gets flushed out of the building. Particles and aerosols generated indoors by the occupants due to breathing and other activities rise with the warm air and are trapped in the hot upper air layer, which is advantageous in hospitals [21]. As we quantify below, the ventilation rate in a naturally ventilated building depends on the strength of the internal heat gains and the sizes of the openings in the building.

# 4. Design guidance for a hospital: natural ventilation

We first calculate the openable area needed to ventilate a space by natural ventilation. The effective openable area $A^*$ in the façade of a building needed to provide a unpolluted lower layer of height $h$ in a room of height $H$ is given by

$$A^* = nC^{3/2} \frac{h^{5/2}}{\sqrt{H-h}},$$ (4.1)

where $n$ is the total number of occupants (we can assume the occupants to be equal strength heat sources), and the empirical constant $C \approx 0.105$ [22,23]. This effective openable area $A^*$ depends on a combination of the total areas $a_t$ and $a_b$ of the top and bottom openings, respectively, given by the relation

$$A^* = \frac{ca_ta_b}{\sqrt{\left(\frac{1}{2}(a_t^2 + a_b^2)\right)}},$$ (4.2)

where $c \approx 0.6$ is a discharge coefficient that accounts for flow contraction and the pressure losses at the openings. For given values of the height of the space $H$ and the number $n$ of occupants, (4.1) and (4.2) can be solved to find the areas $a_t$ and $a_b$ required to ensure the occupants remain in the unpolluted layer of height $h$.

As an example, suppose that a patient, a visitor and, a healthcare worker (i.e. $n = 3$) are in a bay with a total height $H = 4$ m. To ensure the unpolluted layer is higher than 2 m, (4.1) implies that $A^* \approx 0.4$ m$^2$. If the top and bottom areas are equal in size, this requires that they are both approximately 0.64 m$^2$, i.e. they measure, say, 800 by 800 mm. It may be difficult to achieve this amount of opening in practice and so some mechanical extraction may be necessary. We address this possibility in the next section.

When designed properly, the polluted upper layer should be above occupants heads. From (4.1), it is clear that buildings with tall ceilings and with large upper level and lower level openings are optimal for naturally ventilated makeshift hospitals.

# 5. Design guidance for a hospital: mechanically assisted natural ventilation

Buildings are generally equipped with large lower level openings such as windows and doors but often lack large upper level openings. As mentioned above, (4.1) shows that taller spaces are better for natural displacement ventilation—$A^*$ decreases as $H$ increases implying that smaller openings will suffice in a taller space. However, in situations where the required opening area is not available or the space is not tall enough, natural ventilation can be supplemented or replaced by mechanical extraction from the upper part of the space.

In this case, the height $h$ of the lower clean zone is determined by matching the mechanical extraction rate $Q$ with the flow of warm air from the occupants etc. into the upper warm zone. For $n$ occupants, this is given by the formula

$$Q = n^{2/3}CB^{1/3}h^{5/3},$$ (5.1)

where $B = \sum_{i=1}^{n} (W_ig/\rho C_pT)$ the buoyancy flux due to heat fluxes from $n$ sources with heat fluxes $W_i$

watts, $g = 9.81$ m s$^{-2}$ is the acceleration due to gravity and $C_p$ is the specific heat of air. Note now that the height of the space is no longer important and the depth of the clean zone is set by the extraction rate. In principle, this can now be set to any height using a suitable mechanical ventilation rate.

Consider the case of a patient, a visitor and a healthcare worker again, and that each person produces 80 W of heat. The patient lies on a bed at a height 1 m, and to get at a conservative estimate, assume that the visitor and the healthcare worker are at the same level as the patient. In that arrangement, the height $h$ is measured from the height of the bed. Then to attain a clean zone of 2 m, $h = 1$ m, and a total ventilation rate of $Q = 0.405$ m$^3$ s$^{-1}$ is needed. This corresponds to approximately 135 LSP, significantly less than the WHO guideline of 160 LSP mentioned in §2. If the patients and the healthcare workers do not wear masks, a conservative estimate for the flow rate is $Q = 0.523$ m$^3$ s$^{-1}$, or 174 LSP.

# 6. Additional considerations

## 6.1. Attaining and maintaining displacement ventilation

Mixing and displacement ventilation as described above, with completely uniform conditions throughout the space in the former and a two-zone stratification with a clean lower zone and all the contaminants contained in an upper layer in the latter, are at extreme ends of the spectrum. As mentioned before, hot air rises naturally, so even with mixing systems there is often some vertical temperature stratification. Similarly, heat exchange with walls and the floor and other disturbances can cause mixing that brings contaminants down into the 'clean' zone in displacement systems. For example, a person walking can transport contaminant in their wake [24]. Additionally, the flows induced by opening and closing doors can produce undesirable air movement and additional contaminant transport and mixing [25].

## 6.2. Intermediate breathing layer; 'lock-up effect'

Another complicating issue is whether exhaled breath is entrained near the mouth and nose into the body plume and carried to the ceiling or escapes and has its own trajectory in the space. For a person wearing a face covering, the forward momentum is largely lost and usually the exhaled air is immediately captured by the body plume [26]. In the absence of face covering, the exhaled air can escape the body plume and rise under its own buoyancy towards the ceiling. However, this air has less buoyancy than the body plume and so is trapped beneath the contaminated layer, with potentially undesirable consequences. This is known in the literature as the 'lock-up effect' and is often cited as a reason not to use displacement ventilation.

In this case, the exhaled air still rises as a secondary plume and, if it is not immediately entrained into the main body plume, it first settles at an intermediate height and then ultimately is entrained and carried into the upper layer. The correction to the required ventilation rate can be calculated by considering the combined effect of two unequal plumes [27] and is a factor of approximately $1 + (W_{ex}/W)^{1/3}((h - h_M)/(h - h_V))^{5/3}$, where $W_{ex}$ is the heat flux in the exhaled breath, $h_M \sim 1.5$ m is the height of the mouth and $h_V$ is the height of the virtual origin of the body plume. Typically, $W_{ex} \sim 5\% W$, and in the example above where all three persons are at the bed height $h_M \sim 1$ m, $h_V \sim 1$ m, so for an upper layer height of $h = 2.5$ m the increase in the required ventilation rate is a factor of about 37%. The size of this increase emphasizes that the wearing of face coverings which block the forward momentum of the exhaled breath and trap it in the body plume is particularly beneficial.

# 7. Guidelines for displacement ventilation of makeshift hospitals

On the basis of these considerations we suggest the following general guidelines.

— Tall rooms with large upper and lower level openings are suitable for makeshift hospitals.
— Any upper-level ceiling fans or air conditioning units should be removed or not used. Underfloor air distribution should be used if air conditioning is employed [28].
— The lower unpolluted layer height should at least be 2 m deep, and for a building equipped with upper and lower level openings (naturally ventilated), the maximum number of occupants can be calculated by solving (4.1) and (4.2).

— To supplement the unavailability of openings, extractor fans (mounted near the ceiling) could be used. For a given space, the number of occupants, and unpolluted layer height, the extraction flow rate can be calculated from (5.1).

— Except in open-plan hospitals, individual cubicles should be equipped with local lower-level openings. Warm air extract can be centrally arranged.

— In hot climates with uncomfortable outdoor air temperature, cold air should be supplied near the bottom of the room, and hot air exit vents should be made near the ceiling.

# 8. Challenges and future work

Airflow patterns in buildings are dominated by the location of inlets and extracts, and by the air movement induced by people and equipment. In addition to convective flows associated with heat loss from a person, the air movement caused by the wake of a person as they walk can also be significant. These flows are typically orders of magnitude larger than the average ventilation flow, and predicting these patterns is a significant challenge as they depend on these highly variable factors. Current computational models of ventilation flow are unable to capture the full features of these turbulent and often transient flows. On the other hand, since hot air does rise it is possible to develop some general principles as described in this paper, such as the fact that the presence of high-level extraction tends to promote displacement ventilation regardless of the location of inlets. In a naturally ventilated space, this extracted air is vented outside and care must be taken, since contaminant levels are high, that mechanically extracted air is also vented and not re-circulated back into the building.

A further challenge is to model accurately the path of exhaled air and the droplets it contains. Significant progress on the propagation in still air has been made in recent years but the interaction with the heat plume rising from the person and other heat sources such as a computer or other nearby persons is still unclear. It is also important to determine the risk of *inhalation* of infected air and consequently the risk of receiving an infectious viral load. At present, the viral load of SARS-CoV-2 in different size aerosols is unknown as is the dose required for infection. Research into these questions is being conducted as a matter of urgency and we hope that this and the understanding of the importance of ventilation will stand us in better stead this coming winter.

# 9. Conclusion

While adequate ventilation is essential to maintain a healthy indoor environment in hospitals and other public buildings, the dominant flow pattern is a critical factor for the removal of contagions, and cross-contamination between the occupants (patients, healthcare staff and carers). The commonly used mixing ventilation tends to enhance the spread of airborne pathogens throughout the space exposing occupants to possibly infectious airborne particles largely independent of their physical separation from an infected emitter [9]. With the current practice at hospitals, where the staff and patients both wear adequate PPE (masks, respirators, etc.) so that exhaled breath and aerosols are captured by the body plume, displacement ventilation (natural or mechanical) is a suitable and easily implementable alternative to the negative pressure ventilation for makeshift hospitals. Displacement ventilation is economically and environmentally sustainable, and after minor retrofitting, it can easily be implemented into existing buildings.

In order to exit from the current situation and to prevent secondary outbreaks, building ventilation strategy must play an important role. After the end of the lockdown, we believe that public life and practices will change, and persons displaying symptoms will be discouraged from attending public gatherings. If implemented correctly, displacement ventilation in public spaces will provide additional protection from asymptomatic patients, and improve public life. Therefore, we recommend that public spaces (bars, restaurants, supermarkets, etc.), should be retrofitted for displacement ventilation to minimize further spread of the disease after lockdown.

Data accessibility. This article has no additional data.

Authors' contributions. Both authors contributed equally to the content and writing of the paper.

Competing interests. Neither author has any conflict of interest.

Funding. No funding has been received for this article.

Acknowledgements. We are grateful for helpful comments on this article from Sophy Bristow, Megan Davies Wykes, Shiwei Fan and Anna Schroeder. This work was undertaken as a contribution to the Rapid Assistance in Modelling the Pandemic (RAMP) initiative, coordinated by the Royal Society, and was supported by the UK Engineering and

Physical Sciences Research Council (EPSRC) Grand Challenge grant 'Managing Air for Green Inner Cities (MAGIC) grant no. EP/N010221/1.

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
