## [Reviewer comments · Royal Society Open Science]

Review History

RSOS-200680.R0 (Original submission)

Review form: Reviewer 1

Is the manuscript scientifically sound in its present form?

No

Are the interpretations and conclusions justified by the results?

No

Is the language acceptable?

Yes

Do you have any ethical concerns with this paper?

No

Have you any concerns about statistical analyses in this paper?

No

Recommendation?

Reject

Comments to the Author(s)

See the attachment (Appendix A).

Review form: Reviewer 2 (Henry Burridge)**Is the manuscript scientifically sound in its present form?**

Yes

Are the interpretations and conclusions justified by the results?

Yes

Is the language acceptable?

Yes

Do you have any ethical concerns with this paper?

No

Have you any concerns about statistical analyses in this paper?

No

Recommendation?

Accept with minor revision (please list in comments)

Comments to the Author(s)

Please see the attached file (Appendix B).

Decision letter (RSOS-200680.R0)

Dear Dr Bhagat:

I write you in regards to manuscript RSOS-200680 entitled "Displacement ventilation: a viable ventilation strategy for makeshift hospitals and public buildings to contain Covid-19 and other airborne diseases" which you submitted to Royal Society Open Science.

In view of the criticisms of the reviewers and editors, found at the bottom of this letter, your manuscript has been rejected for publication.

Thank you for considering Royal Society Open Science for the publication of your research. I hope this decision will not discourage you from submitting manuscripts in the future.

Yours sincerely,
Royal Society Open Science Editorial Office
Royal Society Open Science
openscience@royalsociety.org

on behalf of Professor Alban Potherat (Associate Editor) and R. Kerry Rowe (Subject Editor)
openscience@royalsociety.org

Editor comments:

The paper is more of a technical advice on ventilation strategy based on existing science. The actual science is already published elsewhere. As such, it does not really qualify as a scientific paper but, perhaps as reviewer 1 recommends, as a magazine paper. Referee 2 makes useful comments that the authors may want to take into account should they chose this route.

Reviewers' Comments to Author:

Reviewer: 1

Comments to the Author(s)

See the attachment.

Reviewer: 2

Comments to the Author(s)

Please see the attached file

Author's Response to Decision Letter for (RSOS-200680.R0)

See Appendices C & D.

RSOS-200680.R1 (Revision)

Review form: Reviewer 1

Is the manuscript scientifically sound in its present form?

Yes

Are the interpretations and conclusions justified by the results?

Yes

Is the language acceptable?

Yes

Do you have any ethical concerns with this paper?

No

Have you any concerns about statistical analyses in this paper?

No

Recommendation?

Accept as is

Comments to the Author(s)

I find the revision satisfactory for "perspective" type article. Thus, I can recommend for the publication.

Review form: Reviewer 2 (Henry Burrige)

Is the manuscript scientifically sound in its present form?

Yes

Are the interpretations and conclusions justified by the results?

Yes

Is the language acceptable?

Yes

Do you have any ethical concerns with this paper?

No

Have you any concerns about statistical analyses in this paper?

No

Recommendation?

Accept as is

Comments to the Author(s)

I thank the authors for their consideration. I'm happy for the article to be published as is. I do note that if it were my choice I would not specify ACE2 receptors (it is, as yet, unclear if they are the only route to infection). Moreover, I note that my comments regarding the possible consideration of two scenarios (in which the level of attendance varied) does not require any analysis of the transients (nor any new analysis), both scenarios have steady-states which (in theory at least) the system would transition between. By considering two (approximately) limiting cases you could show that the interface always formed at a suitable height (or not) in both cases (and by implication for all transients between these cases) - I believe this would be a valuable addition as it shows that human exposure to the (relatively high) concentrations in the upper layer are (or are not) a concern.

Decision letter (RSOS-200680.R1)

Dear Dr Bhagat,

On behalf of the Editors, we are pleased to inform you that your Manuscript RSOS-200680.R1 "Displacement ventilation: a viable ventilation strategy for makeshift hospitals and public buildings to contain Covid-19 and other airborne diseases" has been accepted for publication in Royal Society Open Science subject to minor revision in accordance with the referees' reports. Please find the referees' comments along with any feedback from the Editors below my signature.

Please submit your revised manuscript and required files (see below) no later than 7 days from today's (ie 09-Sep-2020) date. Note: the ScholarOne system will 'lock' if submission of the revision is attempted 7 or more days after the deadline. If you do not think you will be able to meet this deadline please contact the editorial office immediately.

Best regards,

on behalf of Professor Alban Potherat (Associate Editor) and R. Kerry Rowe (Subject Editor)
openscience@royalsociety.org

Associate Editor Comments to Author (Professor Alban Potherat):

Please consider referee suggestions for the final version.

Reviewer comments to Author:

Reviewer: 1
Comments to the Author(s)

I find the revision satisfactory for "perspective" type article. Thus, I can recommend for the publication.

Reviewer: 2
Comments to the Author(s)

I thank the authors for their consideration. I'm happy for the article to be published as is. I do note that if it were my choice I would not specify ACE2 receptors (it is, as yet, unclear if they are the only route to infection). Moreover, I note that my comments regarding the possible consideration of two scenarios (in which the level of attendance varied) does not require any analysis of the transients (nor any new analysis), both scenarios have steady-states which (in theory at least) the system would transition between. By considering two (approximately) limiting cases you could show that the interface always formed at a suitable height (or not) in both cases (and by implication for all transients between these cases) - I believe this would be a valuable addition as it shows that human exposure to the (relatively high) concentrations in the upper layer are (or are not) a concern.

===PREPARING YOUR MANUSCRIPT===

===PREPARING YOUR REVISION IN SCHOLARONE===

- If you are providing image files for potential cover images, please upload these at this step, and inform the editorial office you have done so. You must hold the copyright to any image provided.
- A copy of your point-by-point response to referees and Editors. This will expedite the preparation of your proof.

- Ensure that your data access statement meets the requirements at <https://royalsociety.org/journals/authors/author-guidelines/#data>. You should ensure that you cite the dataset in your reference list. If you have deposited data etc in the Dryad repository, please only include the 'For publication' link at this stage. You should remove the 'For review' link.
- If you are requesting an article processing charge waiver, you must select the relevant waiver option (if requesting a discretionary waiver, the form should have been uploaded at Step 3 'File upload' above).
- If you have uploaded ESM files, please ensure you follow the guidance at <https://royalsociety.org/journals/authors/author-guidelines/#supplementary-material> to include a suitable title and informative caption. An example of appropriate titling and captioning may be found at https://figshare.com/articles/Table_S2_from_Is_there_a_trade-off_between_peak_performance_and_performance_breadth_across_temperatures_for_aerobic_scope_in_teleost_fishes_/3843624.

Author's Response to Decision Letter for (RSOS-200680.R1)

See Appendix E.

Decision letter (RSOS-200680.R2)

Dear Dr Bhagat,

It is a pleasure to accept your manuscript entitled "Displacement ventilation: a viable ventilation strategy for makeshift hospitals and public buildings to contain Covid-19 and other airborne diseases" in its current form for publication in Royal Society Open Science.

COVID-19 rapid publication process:

We are taking steps to expedite the publication of research relevant to the pandemic. If you wish, you can opt to have your paper published as soon as it is ready, rather than waiting for it to be published the scheduled Wednesday.

This means your paper will not be included in the weekly media round-up which the Society sends to journalists ahead of publication. However, it will still appear in the COVID-19

Publishing Collection which journalists will be directed to each week (<https://royalsocietypublishing.org/topic/special-collections/novel-coronavirus-outbreak>).

If you wish to have your paper considered for immediate publication, or to discuss further, please notify openscience_proofs@royalsociety.org and press@royalsociety.org when you respond to this email.

You can expect to receive a proof of your article in the near future. Please contact the production office (openscience_proofs@royalsociety.org) and the editorial office (openscience@royalsociety.org) to let us know if you are likely to be away from e-mail contact -- if you are going to be away, please nominate a co-author (if available) to manage the proofing process, and ensure they are copied into your email to the journal.

on behalf of Professor Alban Potherat (Associate Editor) and R. Kerry Rowe (Subject Editor)
openscience@royalsociety.org

Appendix A

Review Report

Manuscript number: RSOS-200680

Title: Displacement ventilation: a viable ventilation strategy for makeshift hospitals and public buildings to contain Covid-19 and other airborne diseases

Authors: Rajesh K Bhagat and Paul F Linden

Descriptions:

The paper presents design guidance of natural and mechanical displacement ventilation. The authors suggest to use displacement ventilation rather than mixing ventilation for Covid-19 patients. They also suggest using such ventilation in the public spaces at the end of the current lockdown.

Comments:

Although there is nothing wrong nor incorrect with the presented work, there is nothing new on it. Formulation of the effective openable area and the technical extraction rate (presented in section 2(b) and section 3) have already been published by one of the authors. However, the manuscript is well written and provides useful guidance. I suggest publishing this result in a magazine rather than in a scholarly journal. I cannot endorse this manuscript for publication on the Royal Society Open Science.

Appendix B

Review of “Displacement ventilation: a viable ventilation strategy for makeshift hospitals and public buildings to contain Covid-19 and other airborne diseases”

This paper is an excellent contribution to the literature and speedy publication is encouraged. My below comments are relatively minor and should be easy to implement. Moreover, most of the below comments remain optional, i.e. I do not feel that I need to see the paper again unless the authors would like any further input – in which case I would be delighted.

The following is only aspect that I would insist changes... more emphasis needs to be put on the fact that the ventilation flows considered, mixing and displacement, are the two end member cases for ventilation which are very useful when considering the design of ventilation strategies. However, they are only truly realisable in laboratory models and controlled test chambers. Realistic divergences from these design cases must be considered when implementing the chosen strategy. This need to be explicitly mentioned and addressed.

In real buildings there is always more mixing than is considered in the case of pure displacement ventilation (e.g. from draughts, movement within the space, the non-ideal nature of heat sources, time varying nature of heat sources, changes to the geometry of the space like opening of doors/windows/hospital-curtains). I feel that, at the very least, there needs to be acknowledgement of this throughout the paper. Moreover, I would also consider making this a contribution of the paper by having a section “Attaining and maintaining displacement ventilation” or extending section 4, perhaps (I think there is scope for a beneficial short additional section)? This could include: draughts – draughts should be kept to a minimum and where possible these flows should be kept horizontal rather than vertical to minimise disruption of the intended temperature structure within the space; movement within the space – where possible this should be slow and deliberate but where this is in practical the ventilation should be designed to ensure the height of the layer remains well above the movement, e.g. $h \sim 2.5\text{m}$; realistic worst cases for the heat input into the space should be considered in the ventilation design and provision should be made for cases when heat loads will increase (e.g. in winter); within the space, openings (doors, windows, vents, curtains, etc...) should be marked/coloured to highlight whether they have been considered as open in the ventilation design-case, occupants should be educated to keep non-intended openings closed and intentional ventilation openings unobstructed, occupants could be provided with simple charts (graphical or bullet point) on the wall which highlight when and how it might be more safe to change the state of the intended/unintended ventilation openings.

In real buildings, I believe, there is always less mixing than is considered in the case of pure mixing ventilation. I expect because of the complex array of distributed heat sources within most building spaces. Since your focus is on displacement ventilation then you should not feel the need to comment on this unless you wish to do so.

Comments:

Abstract: “\sout{which} exposes” should either have the ‘which’ reinstated or become ‘exposing’.

P2, L18: “from scratch in record time” whilst I like the sentiment, I do not believe this can be rigorously justified, either provide evidence or modify.

P2, L20: “Nightingale-style”, I think it might be worth clarifying what is meant by this.

P2, L23: “in the Central Park” to ‘in Central Park’

P2, L27: Could you please clarify “The virus transmits through direct or indirect human to human contact; fomite and respiratory droplets, respectively”?

P2, L28: “In 2003, a possible SARS-CoV infection to healthcare workers during an aerosol-generating medical procedure had been reported [9]. Furthermore, Yu et al. (2004) presented the evidence of long-distance airborne SARS contagion transport between the apartment blocks in Amoy Gardens housing complex in Hong Kong [10].” These are interesting facts/findings but I think you should make comment on the relevance of each finding in the context of your article.

P2, L35: Replace “meant” with ‘intended’ or ‘designed’.

P2, L55: “and this hot air”, you have not yet mention hot air.

P2, L57+: “In mixing ventilation the occupant is surrounded by air of uniform properties, such as the concentration of a contaminant – a bit like lying in dirty bath water. In contrast, in displacement ventilation the occupant is surrounded by newly arrived air and pollutants generated near or by the person are lifted towards the ceiling by the rising warm air – rather like being in an inverted shower and rinsed with clean water. Of course, for the same amount of contaminant generation and ventilation flow the total amount of contaminant in the space is the same: in mixing ventilation it is everywhere at a relatively low concentration, while in displacement ventilation it is concentrated in a warm layer near the ceiling. When the system is designed correctly this contaminated layer is located above the occupants and so they are only exposed to clean uncontaminated air.” This paper needs to tread a careful balance between simplification for the purposes of broad dissemination of the potential for beneficial displacement ventilation and oversimplification which could lead to poor deployment of the strategy and negative consequences. As a rule, I would suggest that analogies are extremely useful and to be encouraged – however, I am not convinced that the bath and shower analogy is not overly simple. The interactions between pollutants and the flows of different temperature streams of air are far more complex than those of a simple shower. Please review this text, feel free to include the analogy but it could be expressed in terms more like ‘When one takes a bath dirt is removed from one’s body and remains in a relatively dilute but well-mixed state (i.e. relatively uniform concentration) throughout the water in which one lies. When one takes a shower dirt is removed from one’s body and washed away in the flow of water. This is akin to differences between the strategy of mixing ventilation, in which occupants remain surrounded by well-mixed air which contains pollutants – the level of which is diluted by the ventilating flow, and the strategy of displacement ventilation, in which the occupants remain within a lower layer of relatively clean air and pollutants are ‘washed’ away upwards by the ventilating flow.

P3, L23,24: “her” to ‘their’?

P3, L32,33: Feels slightly repetitious.

P3, L41: I do support describing the social distancing rule as “irrelevant”. It remains a ‘rule’ in the case described and is therefore relevant. It may be ‘unsuitable, unsupported, poorly evidenced, etc...’.

P5, L31: “(high density) that” to ‘(high density) air that’.

P5, L53: Given the extreme uncertainty regarding appropriate discharge coefficients of windows/vents/louvres in situ (which need not, indeed should not, be discussed explicitly here) I am surprised that the authors have chosen to specify different values for the top and bottom vents. You could just use $c \sim 0.6$ and simplify (2.2) to $A^* = c \dots$

P5, L58: “(1) implies” is an incorrect/incomplete reference.

P6, (3.1): The document is rightly written with the aim of determining the height of the layer 'h', unlike (2.1), (3.1) could be rewritten to make h the subject, i.e. $h = \dots$. Moreover, shouldn't the buoyancy, $g \Delta T / T$, be rewritten as F/Q ? Doing so would make all of the right-hand side of the equation inputs which can be easily estimated, in fact it might be worth expressing the buoyancy flux F as the heat input W in watts?

The example of the patient and two nurses

I wonder if it is worth putting this example in a separate subsection. Hospitals are dynamic places and I wonder if considering both the case when the patient is unattended, and the patient has two (or more?) visitors might make the example more clearly applicable.

Taking your example in the case of NV when the patient is alone the layer is at a height of 2.78m when the patient is being attended to, the height of the layer slowly decreases to a height of 2m where it remains at a safe height. I believe this highlights that medical staff / visitors remain safe and you could provide an estimate of the time the layer height would take to adjust to the lower based on the filling time scale (which I would expect to be quite long compared to the time a medical team might attend most patients).

In the case of MV, $h = 8.134 * (Q/C)^{3/5} * n^{-2/5} * W^{-1/5}$. In the case that the patient is being visited (taking $W \sim 600W$, $3 * 100W$ for the people and $300W$ for equipment, this I believe is conservative as, e.g. ventilators typically use less than $100W$) then to attain $h=2$, $Q=0.18m^3/s$ (about $50l/s/p$). When the patient is unattended $h=3.4m$. It is worth noting that the change in h is driven primarily by the number of people, n (going from 3 to 1) - which is easily quantifiable whilst h is very insensitive the estimate of the heat/power input into the space - which is perhaps more difficult to quantify. Again, you could include a crude estimate for the time over which the layer height might adjust.

Finally, why not make the example more generic, e.g. two medical staff/visitors, rather than nurses?

Appendix C

Reviewer 1

We thank the reviewer for their very positive comments. Our responses to the detailed points are given below in blue text, as are the changes in the text.

- Abstract: “exposes” should either have the ‘which’ reinstated or become ‘exposing’.
Thanks for pointing this out, We have reinstated it.
- P2, L18: “from scratch in record time” whilst I like the sentiment, I do not believe this can be rigorously justified, either provide evidence or modify.
We have changed the sentence to - healthcare facility/hospital has been built in record time in Wuhan.
- P2, L20: “Nightingale-style”, I think it might be worth clarifying what is meant by this.
In the section Natural ventilation, we write; ‘As was recognised by Florence Nightingale [19], there is evidence that natural ventilation can be beneficial for patient health even when mechanical ventilation is available [1,2], and in many cases it will be the only option.’
- P2, L23: “in the Central Park” to ‘in Central Park’
Corrected
- P2, L27: Could you please clarify “The virus transmits through direct or indirect human to human contact; fomite and respiratory droplets, respectively”?
- P2, L28: “In 2003, a possible SARS-CoV infection to healthcare workers during an aerosol-generating medical procedure had been reported [9]. Furthermore, Yu et al. (2004) presented the evidence of long-distance airborne SARS contagion transport between the apartment blocks in Amoy Gardens housing complex in Hong Kong [10].” These are interesting facts/findings but I think you should make comment on the relevance of each finding in the context of your article.

To address the above two points, we have rewritten the text which now reads – Covid 19 is an infectious disease caused when the SARS-COV-2 virus impacts on ACE2 receptors in the body, usually in the respiratory tract. While the possibility of airborne infection transmission of the virus remains controversial Leung et al. [2020], Morawska and Milton [2020], the virus can remain stable in aerosol form for hours van Doremalen et al. [2020], and can still potentially infect people Remuzzi and Remuzzi [2020], Guo et al. [2020], Chia et al. [2020]. In 2003, a possible SARS-COV infection to healthcare workers during an aerosol-generating medical procedure had been reported Christian et al. [2004]. Furthermore, Yu et al. (2004) presented the evidence of long-distance airborne SARS contagion transport between the apartment blocks in Amoy Gardens housing complex in Hong Kong Yu et al. [2004]. More recently, an outbreak of cases in a restaurant in Guangzhou, China, where CCTV ruled out the possibility of fomite transmission provides further evidence of airborne transmission Lu et al. [2020].

- P2, L35: Replace “meant” with ‘intended’ or ‘designed’.
Replaced
- P2, L55: “and this hot air”, you have not yet mention hot air.
The sentence now reads: ‘Displacement ventilation relies on the fact that inevitably heat is generated within the space, either from the occupants and/or equipment, which produces hot air which, rises and accumulates near the ceiling.’
- P2, L57+: “In mixing ventilation the occupant is surrounded by air of uniform properties, such as the concentration of a contaminant – a bit like lying in dirty bath water. In contrast, in displacement ventilation the occupant is surrounded by newly arrived air and pollutants generated near or by the person are lifted towards the ceiling by the rising warm air – rather like being in an inverted shower and rinsed with clean water. Of course, for the same amount of contaminant generation and ventilation flow the total amount of contaminant in the space is the same: in mixing ventilation it is everywhere at a relatively low concentration, while in displacement ventilation it is concentrated in a warm layer near the ceiling. When the system is designed correctly this contaminated layer is located above the occupants and so they are only exposed to clean uncontaminated air.” This paper needs to tread a careful balance between simplification for the purposes of broad dissemination of the potential for beneficial displacement ventilation and oversimplification which could lead to poor deployment of the strategy and negative consequences. As a rule, I would suggest that analogies are extremely useful and to be encouraged – however, I am not convinced that the bath and shower analogy is not overly simple. The interactions between pollutants and the flows of different

temperature streams of air are far more complex than those of a simple shower. Please review this text, feel free to include the analogy but it could be expressed in terms more like ‘When one takes a bath dirt is removed from one’s body and remains in a relatively dilute but well-mixed state (i.e. relatively uniform concentration) throughout the water in which one lies. When one takes a shower dirt is removed from one’s body and washed away in the flow of water. This is akin to differences between the strategy of mixing ventilation, in which occupants remain surrounded by well-mixed air which contains pollutants – the level of which is diluted by the ventilating flow, and the strategy of displacement ventilation, in which the occupants remain within a lower of relatively clean air and pollutants are ‘washed’ away upwards by the ventilating flow.

We agree with the reviewer that we presented a simplified description, nevertheless, our intention was to highlight the dominating indoor flow pattern, and its importance. The issue of deviations from idealised interior conditions has been discussed, in detail separately. We have modified this text and now write:

‘In mixing ventilation the occupant is surrounded by mixed air of uniform properties, such as the concentration of a contaminant – a bit like lying in dirty bath water. In contrast, in displacement ventilation the occupant is surrounded by newly arrived air and pollutants generated near or by the person are lifted towards the ceiling by the rising warm air – rather like being in an inverted shower and rinsed with clean water. Of course, for the same amount of contaminant generation and ventilation flow the total amount of contaminant in the space is the same: in mixing ventilation it is everywhere at a relatively low concentration, while in displacement ventilation the contaminant is concentrated in a warm layer near the ceiling. When the system is designed correctly this contaminated layer is located above the occupants and so they are only exposed to clean uncontaminated air.’

- P3, L23,24: “her” to ‘their’?
Changed
- P3, L32,33: Feels slightly repetitious.
Here we want to emphasise that droplets produce will be mixed into the environment.
- P3, L41: I do support describing the social distancing rule as “irrelevant”. It remains a ‘rule’ in the case described and is therefore relevant. It may be ‘unsuitable, unsupported, poorly evidenced, etc...’.
We agree and have changed ‘irrelevant’ to ‘problematic’.
- P5, L31: “(high density) that” to ‘(high density) air that’.
Changed
- P5, L53: Given the extreme uncertainty regarding appropriate discharge coefficients of windows/vents/louvres in situ (which need not, indeed should not, be discussed explicitly here) I am surprised that the authors have chosen to specify different values for the top and bottom vents. You could just use $c = 0.6$ and simplify (2.2) to $A^* = c \dots$
We agree and now just use one discharge coefficient $c \approx 0.6$.
- P5, L58: “(1) implies” is an incorrect/incomplete reference.
we have removed the error, thanks.
- P6, (3.1): The document is rightly written with the aim of determining the height of the layer ‘h’, unlike (2.1), (3.1) could be rewritten to make h the subject, i.e. $h = \dots$. Moreover, shouldn’t the buoyancy, $g \Delta T/T$, be rewritten as F/Q ? Doing so would make all of the right-hand side of the equation inputs which can be easily estimated, in fact it might be worth expressing the buoyancy flux F as the heat input W in watts?
We agree that this section was unclear and we have re-written it.
- The example of the patient and two nurses I wonder if it is worth putting this example in a separate subsection. Hospitals are dynamic places and I wonder if considering both the case when the patient is unattended, and the patient has two (or more?) visitors might make the example more clearly applicable. Taking your example in the case of NV when the patient is alone the layer is at a height of 2.78m when the patient is being attended to, the height of the layer slowly decreases to a height of 2m where it remains at a safe height. I believe this highlights that medical staff / visitors remain safe and you could provide an estimate of the time the layer height would take to adjust to the lower based on the filling time scale (which I would expect to be quite long compared to the time a medical team might attend most patients). In the case of MV, $h = 8.134 * (Q/C)^{(3/5)} * n^{(- 2/5)} * W^{(- 1/5)}$. In the case that the patient is being

visited (taking $W = 600W$, $3 \times 100W$ for the people and $300W$ for equipment, this I believe is conservative as, e.g. ventilators typically use less than $100W$) then to attain $h=2$, $Q=0.18m^3/s$ (about $50l/s/p$). When the patient is unattended $h=3.4m$. It is worth noting that the change in h is driven primarily by the number of people, n (going from 3 to 1) - which is easily quantifiable whilst h is very insensitive the estimate of the heat/power input into the space - which is perhaps more difficult to quantify. Again, you could include a crude estimate for the time over which the layer height might adjust. Finally, why not make the example more generic, e.g. two medical staff/visitors, rather than nurses?

These are good suggestions, but for the present purposes we do not wish to consider transient effects. Although we recognise their importance, we prefer to keep the message simple at this stage. However, we have also included a section on the 'Lockup effect', its physical mechanism, and the mitigation strategy. We have changed our example and have made a conservative estimated of the ventilation rate per person. Additionally, we have pointed the issue of the virtual origin of the human plume, and given a conservative estimate of the ventilation rate.

References

- P. Y. Chia, K. K. Coleman, Y. K. Tan, S. W. X. Ong, M. Gum, S. K. Lau, S. Sutjipto, P. H. Lee, B. E. Young, D. K. Milton, et al. Detection of air and surface contamination by severe acute respiratory syndrome coronavirus 2 (sars-cov-2) in hospital rooms of infected patients. medRxiv, 2020.
- M. D. Christian, M. Loutfy, L. C. McDonald, K. F. Martinez, M. Ofner, T. Wong, T. Wallington, W. L. Gold, B. Mederski, K. Green, et al. Possible sars coronavirus transmission during cardiopulmonary resuscitation. Emerging infectious diseases, 10(2):287, 2004.
- Z. Guo, Z. Wang, S. Zhang, X. Li, L. Li, C. Li, Y. Cui, R. Fu, Y. Dong, X. Chi, et al. Aerosol and surface distribution of severe acute respiratory syndrome coronavirus 2 in hospital wards, wuhan, china. Emerging Infectious Diseases, 26(7), 2020.
- N. H. Leung, D. K. Chu, E. Y. Shiu, K.-H. Chan, J. J. McDevitt, B. J. Hau, H.-L. Yen, Y. Li, D. K. Ip, J. M. Peiris, et al. Respiratory virus shedding in exhaled breath and efficacy of face masks. Nature Medicine, pages 1–5, 2020.
- J. Lu, J. Gu, K. Li, C. Xu, W. Su, Z. Lai, D. Zhou, C. Yu, B. Xu, and Z. Yang. Covid-19 outbreak associated with air conditioning in restaurant, guangzhou, china, 2020. Emerging Infectious Diseases, 26(7), 2020.
- L. Morawska and D. K. Milton. It is Time to Address Airborne Transmission of COVID-19. Clinical Infectious Diseases, 07 2020. ISSN 1058-4838. doi: 10.1093/cid/ciaa939. URL <https://doi.org/10.1093/cid/ciaa939>.
- A. Remuzzi and G. Remuzzi. Covid-19 and Italy: what next? The Lancet, 2020.
- N. van Doremalen, T. Bushmaker, D. H. Morris, M. G. Holbrook, A. Gamble, B. N. Williamson, A. Tamin, J. L. Harcourt, N. J. Thornburg, S. I. Gerber, et al. Aerosol and surface stability of sars-cov-2 as compared with sars-cov-1. New England Journal of Medicine, 2020.
- I. T. Yu, Y. Li, T. W. Wong, W. Tam, A. T. Chan, J. H. Lee, D. Y. Leung, and T. Ho. Evidence of airborne transmission of the severe acute respiratory syndrome virus. New England Journal of Medicine, 350(17):1731–1739, 2004.

Appendix D

Reviewer 2

Although there is nothing wrong nor incorrect with the presented work, there is nothing new on it. Formulation of the effective openable area and the technical extraction rate (presented in section 2(b) and section 3) have already been published by one of the authors. However, the manuscript is well written and provides useful guidance. I suggest publishing this result in a magazine rather than in a scholarly journal. I cannot endorse this manuscript for publication on the Royal Society Open Science.

Thank you very much for reading our paper and providing us with you comment but we disagree with your assessment on the novelty of our paper. Although the science is known, it has previously been expressed in the context of the energy saving and thermal comfort that can be achieved by different modes of ventilation. We have reformulated the science to discuss the implications for airborne transmission which is a novel application. In particular, we show through a comparative study, how displacement ventilation can be helpful in containing airborne diseases, particularly in low resource setups. Furthermore, many authors discourage displacement ventilation citing the so-called 'Lockup effect' that arises when the weaker breathing plume separates from the body plume and settles at a lower height below the body plume. To the best of our knowledge the physical mechanisms for this phenomenon are unknown in literature. However, the interaction between a strong and weak plume is known, and we provide a physical mechanism and mitigation strategy. We further disagree that the article should be published in a news magazine; unless the novelty of our main argument, that in displacement ventilation, the contagion produced by an occupant will move to the higher level and extracted, is peer-reviewed, it will not carry sufficient weight to impact policy.

Consequently, we feel that the main new result of our paper, that displacement ventilation prevents mixing of the contagion produced by occupants, along with the engineering solution provided on how to convert a simple building envelope for Covid-19 patients is topical and should be published.

Appendix E

Response to Reviewer 2.

We thank the reviewer for these constructive comments. We agree about receptors and have removed the reference to ACE2.

With regard to the transient effects of changing attendance level, this is not as simple as the referee suggests. There is not always a monotonic change between the two steady states, and the time-dependent behaviour can be quite complicated. A discussion of this is beyond the scope of this paper, as it brings into play a whole range of other time-dependent scenarios, such as changing ventilation rates, and changing occupant positions, which, while interesting, have yet to be properly explored (Bolster et al. 2008).

Reference: Bolster, D.T., Maillard, A. & Linden, P.F. 2008 The response of natural displacement ventilation to time-varying heat sources. *Energy and Buildings*, **40**, 2099-2110.